# Somites are a source of nephron progenitors in zebrafish

Zhenzhen Peng [1], Thitinee Vanichapol[1], Phong Dang Nguyen [2,6], Hao-Han George Chang[1], Katrinka M. Kocha[3], Lori L. O'Brien [4], Peter D. Currie [2,5], Peng Huang [3] & Alan J. Davidson [1] ✉

For over a century it has been believed that the vertebrate kidney arises exclusively from the intermediate mesoderm. Here, we overturn this paradigm by demonstrating that some nephrons, the functional units of the kidney, originate from the somites− blocks of paraxial mesoderm best known for their contribution to muscle and connective tissues. Using a combination of the GESTALT technique to assign developmental ancestry, somite transplantation experiments and Cre-lox fate-mapping, we show that somites can contribute to the nephrons in the adult zebrafish kidney. Our findings uncover an unexpected developmental connection between the somites and kidneys, potentially offering new pathways for developing regenerative treatments for kidney diseases.

Classical embryological studies in the early 20th century have shaped our understanding of vertebrate kidney development, with textbooks consistently portraying the intermediate mesoderm (IM) as the exclusive source of renal progenitors[1–3]. It has been believed that the IM, positioned between the paraxial mesoderm and lateral plate, gives rise to a succession of different kidney types starting with the pronephros (a simple, embryonic kidney), then the mesonephros (transient, functional kidney in some vertebrates), and finally the metanephros, the permanent kidney in amniotes such as birds and mammals[1]. This has led to the widely accepted model that all kidney types, regardless of the vertebrate species, descend sequentially from the IM and within a relatively narrow developmental time window.

In the mouse, the IM forms a relatively dense cord of cells with distinct subsets developing into the nephric duct, mesonephros and metanephros, within a short time period (2−3 days)[1]. By contrast, the zebrafish IM comprises a sparse, narrow strip of cells that appears to differentiate in its entirety into the pronephros[4]. Within 24 h postfertilization (hpf), all the major pronephric cell types have arisen and blood filtration begins by 48 hpf[4,5]. This rapid development likely reflects the physiological need for the zebrafish embryo to

osmoregulate in an aquatic environment outside of the maternal body. The zebrafish mesonephros arises much later, around 10 days postfertilization (dpf), and this has been traced to the abrupt appearance of individual nephron progenitor cells (NPCs)[6,7]. These cells first appear posterior to the swim bladder and then migrate onto the pronephros, cluster together, and proliferate into nephrons[6,7]. The relationship of these NPCs to the IM, which appears to have long since differentiated into the pronephros, presents a conundrum with regard to the current paradigm, and calls into question the lineage relationship between the IM and the mesonephros in teleosts. Indeed, a careful histological analysis of mesonephros formation in Sandkhol Carp larvae suggests that presumptive NPCs first originate near the base of the somites− blocks of paraxial mesoderm best known for their contributions to skeletal muscle, dermis, and vertebrae[8–11].

The somites become compartmentalised into three distinct regions: the myotome, which forms skeletal muscles; the dermomyotome, which generates the dermis and muscle; and the sclerotome, which gives rise to the vertebrae and rib cartilage. There compartments have been extensively studied for their contributions to musculoskeletal development[8–10]. However, lineage tracing studies have

[1]Department of Molecular Medicine & Pathology, The University of Auckland, Auckland, New Zealand. [2]Australian Regenerative Medicine Institute, Monash University, Clayton, Victoria, Australia. [3]Department of Biochemistry and Molecular Biology, Cumming School of Medicine, Alberta Children's Hospital Research Institute, University of Calgary, Calgary, AB, Canada. [4]Department of Cell Biology and Physiology, University of North Carolina at Chapel Hill, Chapel Hill, NC, USA. [5]EMBL Australia, Monash University, Clayton, Victoria, VIC, Australia. [6]Present address: Institut Curie, PSL University, Sorbonne Université, CNRS UMR3215, Inserm U934, Genetics and Developmental Biology, Paris, France. ✉e-mail: a.davidson@auckland.ac.nz

revealed that somites are a diverse source of different progenitor cell types, far exceeding their classical designation as musculoskeletal precursors. It is now appreciated that somitic cells can also give rise to brown adipose tissue[12], endothelial cells[13,14] and fibroblasts[15], indicating a developmental potential much broader than previously appreciated.

Here, by undertaking complementary lineage tracing approaches using GESTALT technology, somite transplantation experiments, and Cre-lox-based genetic fate-mapping, we overturn a century-old developmental paradigm by demonstrating that somitic cells directly contribute to nephron formation in the vertebrate kidney. This discovery challenges the dogma that renal progenitors arise exclusively from the IM and establishes a previously unrecognised developmental pathway with profound implications for our understanding of vertebrate organogenesis.

## Results

### snGESTALT analysis suggests a developmental relationship between muscle and kidney

Consistent with the prior observation in Sandkhol Carp, we found that zebrafish NPCs, which are fluorescently labelled in the Tg(*lhx1a:EGFP*) transgenic line[7,16], first appear at the ventromedial border of the anteriormost somites, adjacent to the pronephric tubules (Fig. 1a). From here they migrate and expand in number around the axial vessels (Supplementary Fig. 1a, b)[7].

To explore the hypothesis that these NPCs originate from the somites, we performed a lineage mapping analysis using a modified version of the scGESTALT (single cell transcriptome and genome editing of synthetic target arrays for lineage tracing) technique[17]. This approach maps developmental relationships between different lineages of cells using a combination of genetic barcode editing by CRISPR/Cas9 and single cell RNA-Sequencing. Transgenic zebrafish carrying the heat shock inducible GESTALT barcode (sites 1-9) were crossed to a heat shock-inducible Cas9 line (constitutively expressing guide RNAs to sites 5-9) and injected at the single-cell stage with Cas9 protein and guide RNAs 1-4. This ensures editing occurs at barcode sites 1-4 during the pre-gastrulation stages, capturing the earliest lineage decisions (Fig. 1b). The embryos were then heat shocked at the 12-somite stage (15 hpf) to induce a second round of editing at sites 5-9 during mid-somitogenesis (Fig. 1b). This dual editing strategy allows us to capture both early lineage bifurcations and later developmental decisions, enabling reconstruction of more detailed lineage trees. Larvae with successful editing at both early and late barcode sites (Supplementary Fig. 1c) were raised to 1.5 months of age (~1 cm Standard Length), the kidney and muscle were isolated, together with the brain as a control. Rather than performing scRNA-Seq, we chose instead to undertake single nuclei RNA-Seq, due to the multinuclear nature of some muscle fibres (herein we refer to the analysis as single nuclear (sn) GESTALT). Therefore, the nuclei were purified from each tissue, the brain and muscle samples were pooled together, and the nuclei were sorted and encapsulated using the 10X Chromium platform. We recovered the transcriptomes from four brain/muscle samples and three kidney samples. Using the unsupervised modularity-based clustering approach (UMAP), nuclei were grouped into clusters. 31 transcriptionally distinct muscle and brain populations (Supplementary Fig. 1d, e) and 18 transcriptionally-distinct kidney populations (Supplementary Fig. 1i) were resolved. Established anchor markers reported in the literature and the ZFIN database were then used to assign presumptive cell identities to each population. Clusters expressing *Titin* genes (*ttn.1*, *ttn.2*) and myosin heavy chain genes (*myhc4*) were classified as muscle nuclei (*n* = 14,364), and clusters expressing neuronal genes (*luzp2*, *nrxn3a* and *grid1b*) were classified as brain nuclei (*n* = 10,269) (Supplementary Fig. 1f). This resulted in the identification of 12 muscle and 9 brain populations. In addition, we found presumptive populations of immune (*ptprc*), endothelial (*egfl7*), blood (*hbaa1*) and mural cells (*myh11a*) (Supplementary Fig. 1h). Three

populations (cluster 12, 19 and 24) could not be assigned to either muscle or brain, due to a lack of expression of anchor genes.

To further classify the muscle clusters, we compared the differentially expressed genes in each cluster with markers specific to muscle cell types (fast and slow), as well as developmental stage (Supplementary data 1)[18–22]. From this we identified 6 distinct fast muscle (FM) subtypes (FM 1-6 in Fig. 1c, d). FM1 and FM2 show enriched expression of the parvalbumin genes, *pvalb1* and *pvalb2*, which play a role in fish muscle relaxation[23]. FM3 to FM6 show high expression of the ryanodine receptor genes, *ryr1b* and *ryr3*, *mef2cb*, *msi2b* and *sox6*, which are markers of developing fast muscle fibers (Fig. 1d and Supplementary Fig. 1g)[24]. Given that FM1 and FM2 express lower levels of the pro-differentiation genes, we speculate that these clusters correspond to more mature fast muscle fibers cells. We identified one slow muscle fiber population (SM) that expresses markers such as *smyhc2*, *myh7bb*, *mybpc1* and *tnnt2e*[18] (Fig. 1d). In addition, we found a mixed fast/slow population that expresses *ryr1b* and *ryr3*, but also the slow muscle fiber gene *ryr1a*, suggestive of common progenitor population of intermediate fibres[10] (Fig. 1d). Two mixed fibroblast populations that expressed several collagen genes were also identified (Fig. 1d).

For the brain nuclei, we grouped them into 9 transcriptionally-distinct neuron populations that included 5 glutamatergic subtypes (expressing *gria2b*, *gria3b* and *epha6*), one *fat2*+ subtype and 2 undefined subtypes (Fig. 1c, e and Supplementary data 2). All of these populations, except for glutamatergic neuron-5, are likely mature neurons, as they expressed the markers *rbfox3a* and *map2*[25,26]. Our analysis also identified a presumptive group of neuronal stem cells, radial glia cells (expressing *s100b*, *atp1a1b* and *msi1*), and oligodendrocytes (expressing *flj13639*, *mpz* and *mbpa*; Fig. 1c, e).

Nuclei from the kidney were grouped into populations corresponding to the nephron segments (7 subtypes, *n* = 3854 nuclei), haematopoietic cell lineages (6 subtypes, *n* = 2939 nuclei) and neurons (2 subtypes, *n* = 214 nuclei) (Fig. 1f, Supplementary Fig. 1j, k). These clusters were annotated with anchor genes, as well as differentially expressed genes[6,7,27] (Supplementary data 3). While no podocytes or NPC clusters were identified, we did resolve proximal and distal tubule nephron populations. These comprised one proximal convoluted tubule population (PCT), which was defined by expression of *lrp2a*, *slc20a1a* and *slc34a1a*[28–30], and two proximal straight tubule populations (PST1 and PST2), based on the expression of *trpm7* and *slc13a1*[29–31] (Fig. 1g). Differential gene expression analysis between these subtypes revealed that PST2 also expresses *rap1gap*, *slc6a6a* and *slc12a9*. For the distal nephron segments, we identified distal early (DE) cells, which express *clcnk* and *slc12a1*, and distal late (DL) cells that express *wnt9a* and *slc12a3*[30–32]. We also have identified a new DL-like population that specifically expresses *atp1a1a.3* and *scl9a3.2* (Fig. 1h). GO Term analysis for this population is consistent with a role in ion transport (Supplementary Fig. 1l). To determine the spatial location of this DL-like segment we performed in situ hybridisation on adult kidney sections and found expression in the major collecting ducts (running down the midline of the kidney) and in the distal-most portion of nephrons that fuse with the collecting ducts (Fig. 1i, j).

To determine the relationship of cells from kidney and muscle using snGESTALT, we prepared and sequenced the snGESTALT lineage libraries, resulting in barcode recovery of 358 nuclei from two juvenile zebrafish, respectively, corresponding to ~2% of all profiled cells. We filtered out the barcodes with only one UMI (Unique Molecular Identifiers) and constructed lineage trees for the remaining barcodes using the published maximum parsimony approach[17], anchoring the tree with edits at sites 1-4 and extended it with edits at sites 5-9. Although the recovery rate of the barcodes was low, most likely due to weak expression of the heat-shock promoter and lower recovery from nuclei, they cover the majority of the identified populations in all three tissues (Supplementary Fig. 2a, b). A shared barcode between different

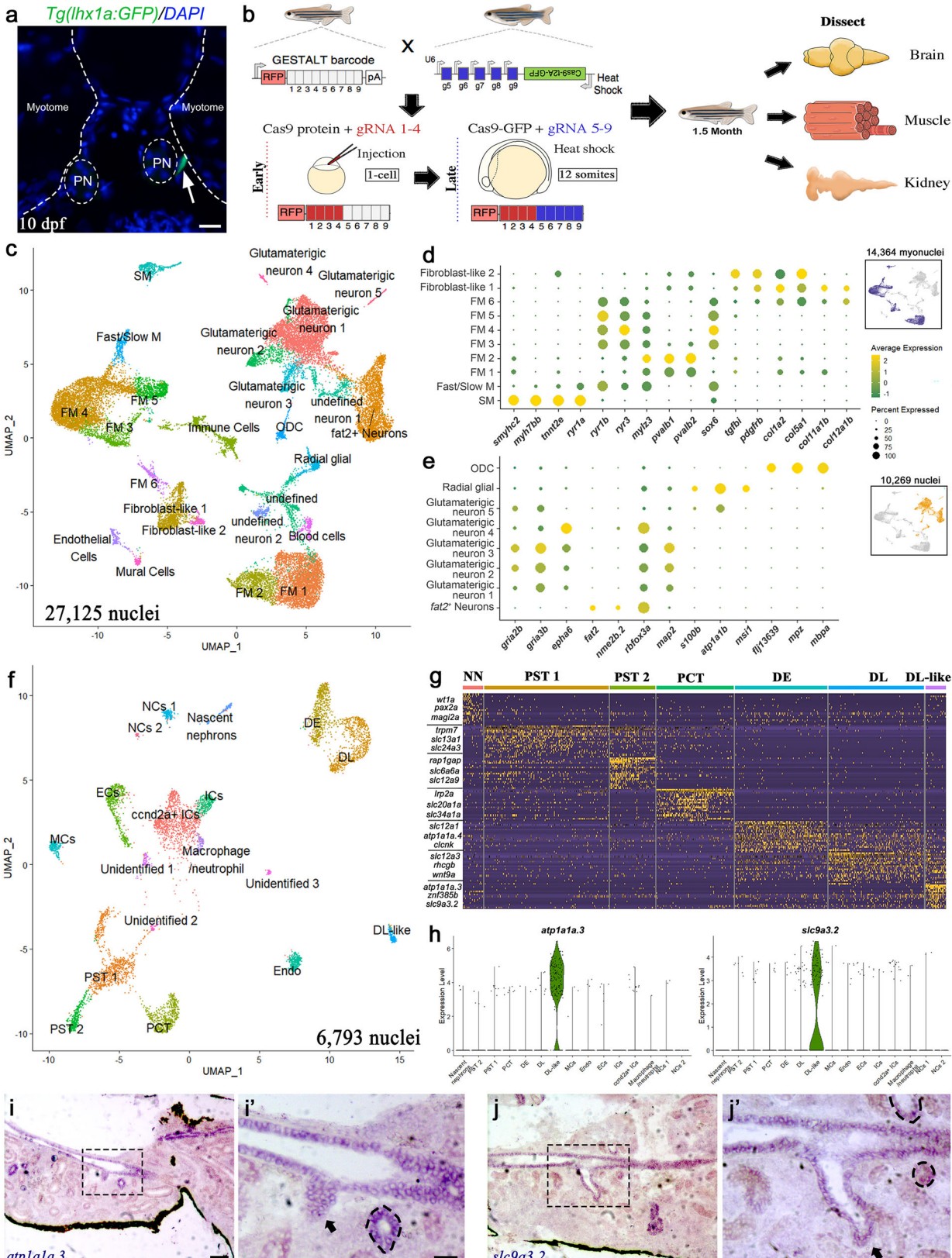

types of cells indicates a shared ancestry and we found that the majority of these were restricted to single tissue types e.g. between different types of muscle cells (Clade a branch a3, Clade c, Fig. 2a; Clade a branch a1, a2 and Clade c. Figure 2b) or between nephron segments (Clade a branch a1, Clade b, Fig. 2a) or different neuronal cells (Clade d, Supplementary Fig. 2a). We found one barcode shared

by a fast muscle cell (FM5) and a *fat2*⁺ neuronal cell, likely reflecting the lineage relationship between bipotent neural mesodermal progenitors that contribute to the spinal cord and somitic mesoderm[33] (Clade a branch a3 in Supplementary Fig. 2b). No shared barcodes were found between the kidney and brain (Supplementary Fig. 2). However, we found two examples of barcode sharing between the muscle and

**Fig. 1 | snRNA-Seq identifies various cell types from zebrafish muscle, kidney, and brain. a** Cross-section of 10 dpf zebrafish larvae showing the *lhx1a*-GFP⁺ kidney progenitor cells located at the interstitial region between the myotome and pronephros (PN) (scale bar, 20 μm). Representative image from one of three biologically independent larvae (*n* = 3). **b** Schematics showing GESTALT barcode editing and tissue harvest protocol for transcriptomes and lineage recordings analysis. **c** UMAP of 27,125 nuclei clustered into muscle and brain cell types from four biologically independent fish (*n* = 4). M Muscle; FM fast muscle, SM Slow muscle, ODC oligodendrocyte. **d, e** Dotplot of gene expression patterns of select marker genes (rows) for identified subcluster (columns) from muscle (n = 14,364 myonuclei) and brain (*n* = 10,269 nuclei). Dot size represents the percentage of nuclei expressing the marker; color represents the average scaled expression level. Inset highlights (blue and orange) the muscle and brain clusters in (**c**). **f** UMAP of 6793 nuclei clustered into kidney cell types from three biologically independent fish (*n* = 3). EC

endothelial cells, MC Mural cells, IC Immune cells, NC kidney neuron cells, PST1 proximal straight tubule 1, PST2 proximal straight tubule 2, PCT Proximal convoluted tubule; DE Distal early tubule; DL Distal late tubule. **g** Heat map of scaled gene expression of top 20 differentially expressed genes from each nephron segment (Supplementary data 4). Key marker genes for each nephron segment are indicated on the left. NN nascent nephrons. **h** Violin plot of specific markers of the DL-like tubular population. **i** Expression of *atp1a1a.3* in kidney cross-sections (scale bar, 100 μm). **i'** Close-up image of black-dashed box in (**i**), black-dotted circle demarcates mesonephric tubule expressing *atp1a1a.3* (scale bar, 20 μm), Representative image from one of three biologically independent fish kidneys (*n* = 3). **j** Expression of *slc9a3.2* in kidney cross-sections. **j'** Close-up image of black-dashed box in (**j**), black-dotted circles demarcate mesonephric tubule expressing *slc9a3.2*, Representative image from one of three biologically independent fish kidneys (*n* = 3).

kidney lineages. The first of these revealed a lineage relationship between an FM3 cell and a DL-like tubular cell (Fish 1 Clade a branch a2, Fig. 2a), while the other linked an FM4 cell to three tubular cells (PST1, PST2 and DL, Fish 2 Clade b, Fig. 2b). These findings are consistent with the hypothesis that the somites can give rise to kidney tissue.

### Transplantation experiments identify somite 3 as a source of RPCs

To further test whether somites are a source of NPCs, we performed somite transplantation experiments. We selected the 3rd somite, based on its proximity to the where the first NPCs arise later in development, and dissected this from transgenic embryos ubiquitously expressing *egfp* at either the 12- or 18-somite stages (~15 hpf or 18 hpf, respectively). These donor somites were then transplanted into equivalently staged unlabelled recipients where the corresponding somite was removed (Fig. 3a and see Supplementary Fig. 3a–e for an overview of the transplant procedure). In surviving recipients, we observed successful engraftment through the presence of GFP⁺ donor-derived muscle fibres at the transplantation site (Fig. 3b). Follow-up analysis at 1.5 months revealed continued donor-derived contribution to muscle fibres at the transplantation site as well as donor-derived Hnf1b⁺ mesonephric kidney tubules and Podxl⁺ glomeruli (a transmembrane sialomucin expressed on the surface of podocytes[34]) in the mesonephros (*n* = 5/100 recipients, with *n* = 3 with 12-somite stage donors and *n* = 2 from 18-somite stage donors; Fig. 3c, d and Supplementary Fig. 3f–h). Notably, in one recipient, donor-derived tubules were found contralateral to the transplantation side, consistent with our previous observations that NPCs exhibit migratory behaviour[6] (Fig. 3c'). Despite the somite being easily separatable from the adjacent IM, we explored the potential for contaminating IM cells to be co-isolated. To do this, we performed somite transplantations with Tg(*cdh17:EGFP*) donors, which is expressed by the IM/pronephric tubules, but found no donor-derived GFP⁺ kidney cells in the recipient fish at 4 dpf (*n* = 23; Supplementary Fig. 3i, j).

### Genetic fate-mapping reveals that NPCs are derived from *nkx3.1* positive progenitors

To provide further evidence supporting a renal-somite lineage relationship, we conducted conventional 4-hydroxytamoxifen (4-OHT)-inducible Cre-mediated (CreERT2) genetic fate-mapping experiments in vivo using transgenic lines employing the paraxial mesoderm-specific *mesogenin1* (*msgn1*) promoter[13] and a GAL4-UAS system[35] involving the sclerotome marker *NK3 homeobox 1* (*nkx3.1*)[15] (Fig. 4a). In *msgn1:Cre-ERT2; βactin:Switch* fish, paraxial mesoderm-derived cells were induced to express mCherry with 4-OHT treatment at the 50% epiboly stage (5.25 hpf). Similarly, in *nkx3.1:Gal4; UAS:Cre-ERT2; ubi:Switch* fish, *nkx3.1*-expressing sclerotome cells were targeted for Cre-mediated recombination at 24 hpf (Fig. 4b). This approach enabled us to track and compare the fate of cells derived from the pan-paraxial mesoderm (*msgn1*) at early stages and the sclerotome subpopulation

(*nkx3.1*) at later stages. Both lines were treated with 4-OHT for a duration of 24 h. As expected, the mCherry⁺ cells derived from the *msgn1* lineage gave rise to muscle fibres at 48 hpf (Supplementary Fig. 4b, c), while no labelling of the IM-derived pronephros was observed at either 48 hpf or at 14 day post-fertilisation (corresponding to the 7 mm stage; Supplementary Fig. 4d and Fig. 4c, d, *n* = 5). The mCherry⁺ cells derived from the *nkx3.1* lineage were found associated with the myotendinous junction, centrum, neural arch and hemal arch (as reported by Ma et al. 2023, Supplementary Fig. 4f). Consistently, some mCherry⁺ muscle fibres were also observed in the ventral part of myotome (Supplementary Fig. 4f, yellow arrow heads). Embryos of both lines without 4-OHT induced Cre showed no presence of mCherry⁺ cells (Supplementary Fig. 4e, f). We examined the recombined animals from both lines at the 7 mm stage, a crucial period when the mesonephric kidney begins to form. We assessed expression of *pax2a*, a transcription factor marking NPCs and early renal epithelial cells, and *hnf1b*, which is essential for nephron segmentation and tubule maturation[31]. We observed the contribution of mCherry⁺ cells to nascent Pax2⁺ nephrons (*n* = 5; Fig. 4c) and NPC clusters (*n* = 5; Fig. 4e). Additionally, we observed contribution of mCherry⁺ cells to Hnf1b⁺ nephrons in both lines (*n* = 10; Fig. 4d, f), including mCherry⁺ glomeruli (Fig. 4f, *n* = 10). Recombined animals (1.5 months of age) of both lines exhibited mCherry labelling in nephrons scattered throughout the kidney with mosaic contribution in the glomeruli, proximal (LTL⁺, Lotus Tetragonolobus Lectin) and distal tubules (DBA⁺, Dolichos Biflorus Agglutinin) (Fig. 4g, h, i, *n* = 10 per line). In addition to the nephron segments, we have also observed contribution to stromal-like cells in the kidney. However, in the absence of validated stromal markers in zebrafish, the identities of these cells are unclear (Supplementary Fig. 4i). Additionally, 4-OHT was administered to *msgn1* embryos at the later stages of 3- and 10-somites, where there is no overlap in expression between *msgn1* and the IM marker *pax2a* (Supplementary Fig. 4g). A similar efficiency of nephron contribution was found among the three different 4-OHT treatment stages (Fig. 4o), although the labeling of muscle fibers was more restricted to the posterior somites in the 10-somite treated animals compared to the earlier treatment windows (Supplementary Fig. 4h, Fig. 4m). The *nkx3.1* line demonstrated 6-fold greater efficiency in contributing to nephron formation compared to *msgn1* line (Fig. 4n, o). Taken together, the data from both *msgn1* and *nkx3.1* studies provide further independent evidence for a somite origin of NPCs.

## Discussion

In this study we demonstrate through three independent lineage tracing methods that somites serve as a source of NPCs for mesonephric nephrons in zebrafish. This finding challenges the prevailing assumption that the IM is the exclusive origin of all kidney cell types in vertebrate embryos. This long-standing belief primarily emerged from studies of chick and mammalian embryogenesis, where the IM develops progressively as the embryonic axis elongates and forms

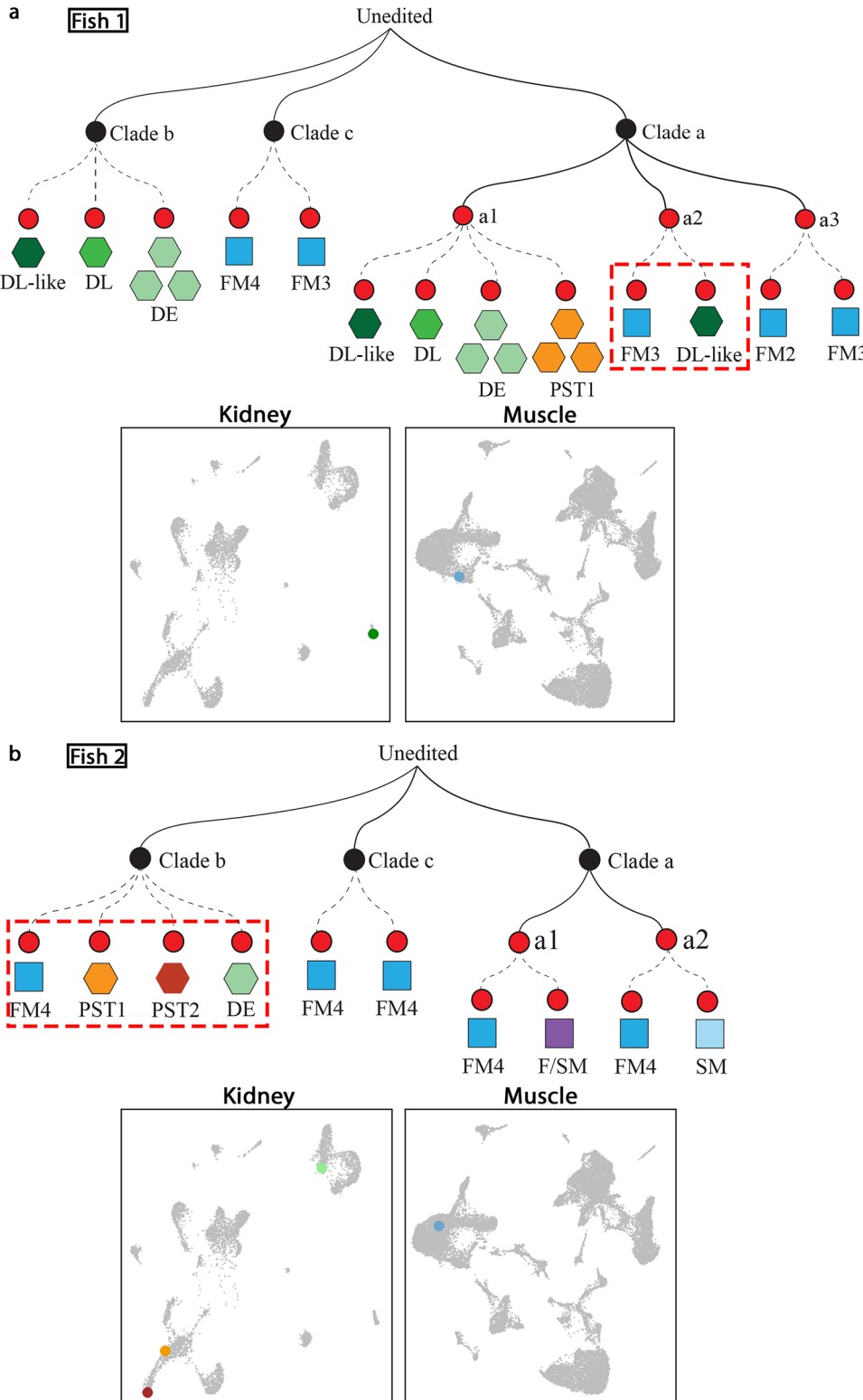

**Fig. 2 | Reconstructed lineage trees reveal linkage between muscle and kidney cells in juvenile zebrafish. a** Mini-tree showing lineage branches of kidney and muscle cells from Fish 1 that share the same barcode (selected branches from Supplementary Fig. 2a). **b** Mini-tree showing lineage branches of kidney and muscle cells from Fish 2 that share the same barcode (selected branches from Supplementary Fig. 2b). Black nodes represent early edits at barcode sites 1–4; red nodes represent late edits at sites 5–9. Solid black lines indicate inferred lineage relationships between nodes. Dashed lines connect individual cells to shared barcodes. Each shape represents a single cell, color-coded by cell type. Bottom panels show UMAP plots highlighting the corresponding kidney and muscle cells that share the same barcodes (red dashed boxes). PST proximal straight tubule; DE distal early tubule, DL distal late tubule, FM fast muscle, SM slow muscle, F/SM fast/slow muscle.

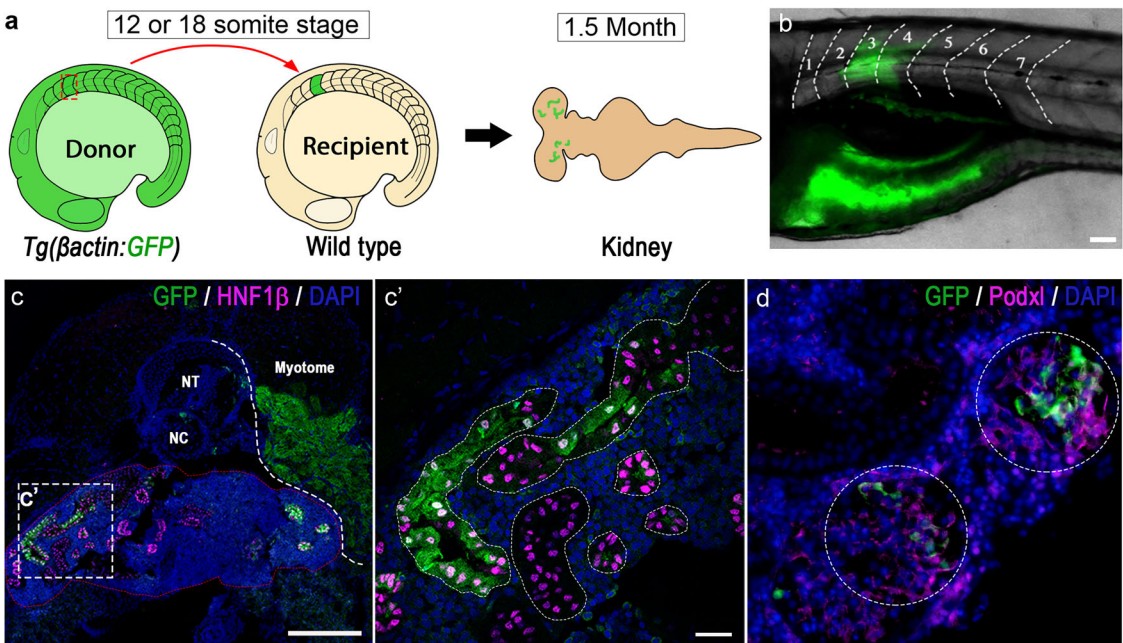

**Fig. 3 | Transplanted GFP⁺ somite contributed to mesonephric nephrons in recipient fish. a** Schematic of somite transplantation showing a GFP⁺ donor somite is harvested at 12 (15 hpf) or 18 (18 hpf) somite stage and transplanted into a wild-type recipient embryo at equivalent developmental stage, and kidney of the recipient fish is examined at 1.5-month of the age. **b** Image of a recipient larvae (6 dpf) developed donor derived GFP⁺ muscle fibers at the transplantation site (somite 3, transplanted at 12 somite stage). Scale bar, 20 μm. **c** Cross section of same recipient juvenile fish (at 1.5-month stage) immunostained for GFP and renal tubular marker Hnf1b, showing donor somite contributed tubular structures (Scale bar, 100 μm). **c'** Close-up image of the area boxed in c showing Hnf1b⁺ mesonephric tubules containing donor somite derived GFP⁺ cells (Scale bar, 20 μm). Similar results were observed in 3 biologically independent recipients with positive contribution (*n* = 3). **d** Immunostaining of donor derived Podxl⁺/GFP⁺ glomeruli, Scale bar, 20 μm. Similar results were observed in 3 biologically independent recipients with positive contribution (*n* = 3). For merged images, levels adjustments were applied independently to individual colour channels for clarity. All adjustments were applied equally across the entire image.

nephrogenic cords alongside the somites. These IM cells are the source of the sequentially-arising kidney types— the pronephros and mesonephros in the rostral trunk and the metanephros in the caudal trunk of mammalian and chick embryos[3,36,37]. By contrast, zebrafish kidney development follows a markedly different trajectory. The IM forms in its entirety by early somitogenesis stages and then appears to differentiate *in toto* into the pronephros, with no evidence of 'remnant' nephrogenic cord cells remaining to serve as a source for the mesonephros. Unlike the other models, the zebrafish mesonephros emerges much later in development, in more mature, free-swimming larva. The absence of persisting nephrogenic cord cells in zebrafish has long been a developmental mystery that our work now resolves by revealing that the somites are an alternative source of NPCs.

During zebrafish somitogenesis, the somite becomes compartmentalised into discrete domains that include the primary myotome (contributing to both fast and slow muscle fibres)[10], a dermomyotome-like external cell layer, and dorsal and ventral sclerotome populations that are a source of multiple types of mesenchymal cells and the vertebrae[15,38,39]. Our lineage tracing with *msgn1:Cre-ERT2* does not resolve which of these compartments is the source of NPCs, as *msgn* is expressed from early gastrulation onwards throughout the paraxial mesoderm[40]. By contrast, *nkx3.1* is more restricted in the somite and has a later onset of expression, where it predominantly labels sclerotome progenitors[15,41]. This result, coupled to the observations that the first NPCs arise near the base of the myotome, suggest that ventral sclerotome population may be the source of NPCs. Whether a subset of sclerotome progenitors persist to ~10 dpf, when the first NPCs arise, or whether there is an intermediary cell type/s involved in the developmental pathway from somite to NPC is not clear but can be addressed in the future with additional lineage-restricted Cre-lox tools.

Alternatively, as *nkx3.1-cre* also labels some muscle fibres[15], together with our GESTALT analysis where we found that some fast muscle

cells share a common ancestry with kidney cells, we cannot rule out that NPCs descend from the myotome compartment. Although the trunk muscle initially arises from the primary myotome, during the larval and adult stages it is greatly expanded by the differentiation of muscle stem cells on the myotome surface[10]. It is possible that these muscle stem cells are the source of NPCs, and this would fit with the observations that NPCs arise during larval stages when secondary myogenesis is active, together with their first appearance at the surface of the myotome. In line with this, the snGESTALT results found that the muscle cells sharing a common ancestry with the renal cells have an immature transcriptional profile, suggestive of being newly-derived during secondary myogenesis[10]. A link between muscle stem cells and the kidney is attractive as both tissues show lifelong indeterminant hyperplasia in teleosts[7,10] and a common stem cell population may help coordinate this growth.

While our results link the somites to the mesonephric kidney, we cannot completely rule-out a dual contribution that also includes the IM. At present, we do not favour such a model as the temporal gap between when the IM forms (~10 hpf) and NPCs appearing 10 days later is a substantial length of time for these cells to persist. We have conducted extensive gene expression mapping of NPC markers, including *pax2a*, *wt1b*, *lhx1a*, and have not seen any evidence of 'remnant' IM[6]. It is possible that some IM cells differentiate into an intermediary cell type that does not express traditional markers of NPCs yet retains the potential to adopt an NPC fate later in development.

Our findings raise the question of whether the somite-to-kidney developmental pathway is a unique adaptation found in teleosts or whether it reflects an ancient vertebrate developmental mechanism that has been overlooked in mammals and birds. Lineage labelling studies in chick embryos have shown that the paraxial mesoderm contributes to the mesonephros and metanephros but is restricted to

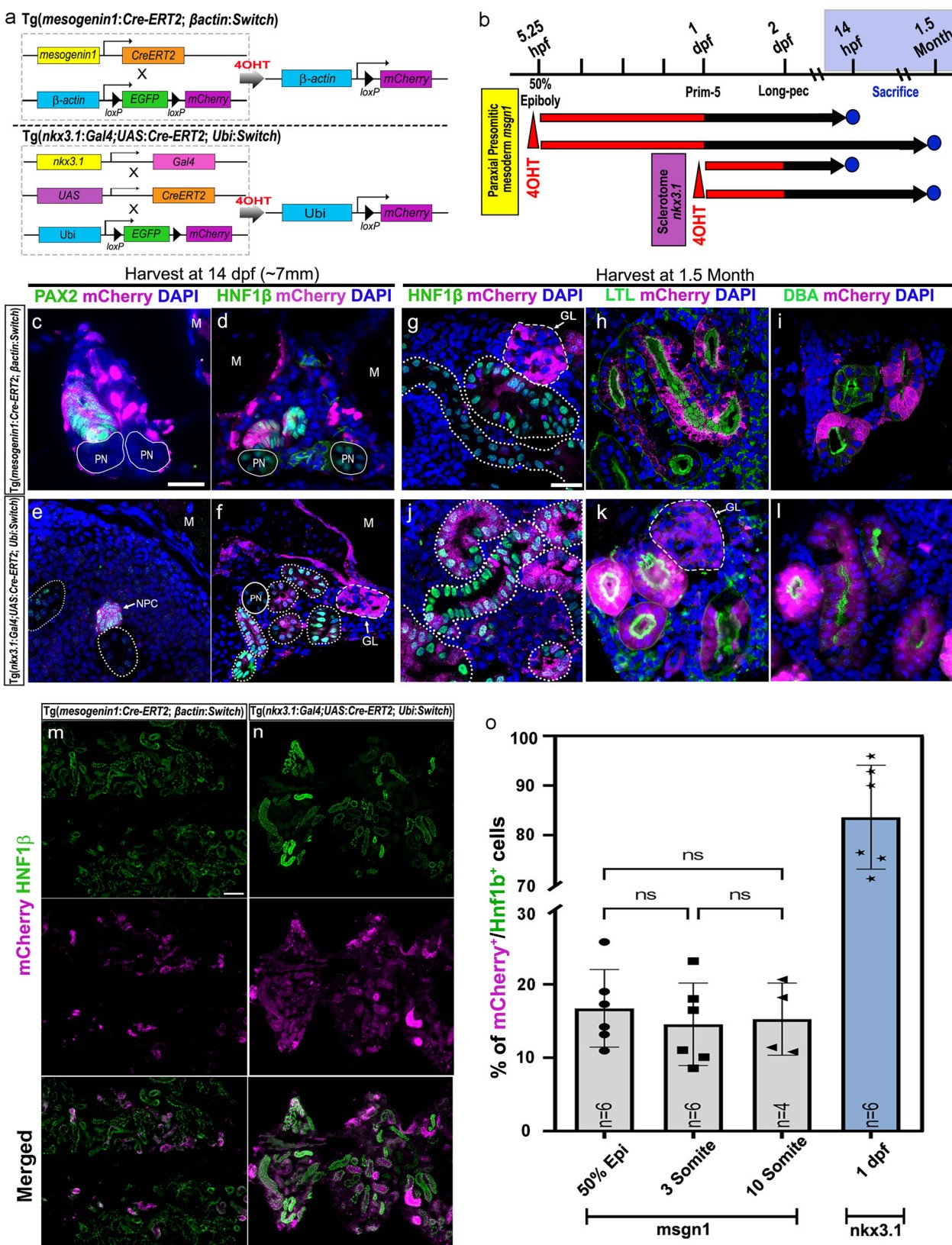

stromal cells[42]. While the stromal and NPC populations were initially considered to be developmentally distinct lineages, more recent observations suggest a much closer relationship. Lineage tracing of mouse stromal precursors using *Foxd1-cre*, which shows earlier expression in the somites, labels the metanephric stroma but also small populations of metanephric NPCs[43,44]. Moreover, knockout of the

*Pax2* transcription factor in metanephric nephron progenitors causes them to transdifferentiate into stromal progenitors[45]. In the developing human metanephric kidney, the transcriptional profiles of nephron and stromal progenitors overlap substantially, including shared expression of *FOXD1*, considered a specific stromal marker in mouse[46]. Together, these observations raise a provocative hypothesis that the

**Fig. 4 | Cre-loxP based lineage-tracing techniques unveil a somitic origin for zebrafish mesonephric nephrons. a** Schematic representation showing the cross between the *msgn1*:Cre-ERT2 line and *bactin*:Switch (loxP-EGFP-loxP-mCherry) line, and the *nkx3.1*:Gal4;UAS:Cre-ERT2 line and *Ubi*:Switch line, followed by treatment with 10 μM 4-OHT to activate Cre recombinase and induce mCherry expression in targeting cells. **b** Experimental timeline of 4-OHT treatment in both transgenic lines. **c–f** Cross-sections of 4-OHT treated 7 mm larvae immunostained for renal markers (Pax2, Hnf1β) and mCherry⁺ cells. Tg(*msgn1*:Cre-ERT2;*bactin*:Switch) (**c**, **d**) and Tg(*nkx3.1:Gal4; UAS:Cre-ERT2; ubi:Switch*) (**e**, **f**), (Scale bar, 20 μm); PN: pronephros; M: muscle, NPC: Nephron progenitor clusters; GL: Glomerulus. Images are representative of five biologically independent larvae per transgenic line (*n* = 5 per line), with similar results. **g–l** Sections of 4-OHT-treated 1.5-month-old zebrafish kidneys immunostained for renal markers: HNF1β (white dotted outlines), Lotus tetragonolobus lectin (LTL), Dolichos biflorus agglutinin (DBA), and mCherry⁺ cells. Images show donor-derived contributions to the glomerulus (GL; white dashed outlines), proximal tubule (LTL), and distal tubule (DBA) of the nephron. PN, pronephros (solid white outlines). Scale bar, 20 μm.

Images are representative of ten biologically independent fish per transgenic line (*n* = 10 per line), with similar results. **m**, **n** Representative low-magnification images of kidney sections from both transgenic lines for quantification. Scale bar, 100 μm. **o** Quantification of the percentage of mCherry⁺/Hnf1b⁺ cells in 1.5-month Tg(*msgn1*:Cre-ERT2;*bactin*:Switch) and Tg(*nkx3.1:Gal4; UAS:Cre-ERT2; ubi:Switch*) fish kidneys. For the *msgn1*-Cre-ERT2 line, recombination occurred at 50% epiboly, 3-somite, and 10-somite stages, and for the *nkx3.1*-Gal4-UAS-Cre line, recombination occurred at 1-day post-fertilization (1dpf). Data points correspond to the mean percentage of mCherry⁺/Hnf1b⁺ cells ± SEM. Each dataset consists of three sections quantified per fish, with each section 14 μm in thickness, and every fifth section was quantified. Statistical significance was assessed using a two-sided one-way ANOVA followed by Tukey's post-hoc test; ***$P < 0.001$. For the 50% epiboly and 3-somite stages, *n* = 6 fish per time point, for the 10-somite stage, *n* = 4, and for the 1dpf stage, *n* = 6. Source data are provided as a Source Data file. For all merged images, levels adjustments were applied independently to individual colour channels for clarity. All adjustments were applied equally across the entire image.

somite-to-kidney pathway we describe in zebrafish may also exist in mammals and birds but it has become co-opted to largely generate stromal, rather than nephron, progenitors.

In conclusion, our work challenges the traditional view of the IM as the sole source of NPCs. It extends the remarkable plasticity of somites as a key developmental reservoir of tissue progenitors and offers a provocative lens through which to reconsider amniote kidney development. Our research opens the possibility of inducing human nephron progenitors in vitro by manipulating a somite-to-kidney pathway and utilising them in regenerative therapies aimed at treating kidney disease.

## Methods

### Zebrafish husbandry
Zebrafish husbandry and manipulations were conducted at the facilities of the University of Auckland, Faculty of Medical and Health Sciences. This study was approved by the University of Auckland Animal Ethics Committee under approved protocol AEC22634.

### Zebrafish transgenic lines
Zebrafish (*Danio rerio*) were maintained as previously described[47] and in accordance with Institutional Animal Care and Use committee protocols. Zebrafish of mixed sexes, age under 30 days post-fertilization (dpf) or 1.5 months, were used in experiments. All lines were maintained on an AB genetic background. The following transgenic lines were used: Tg(*lhx1a:EGFP*) from Prof. Hukriede's lab, University of Pittsburgh; Tg(*hsp70l:Cas9-t2A-GFP, 5xU6:sgRNA*) and Tg(*hspDRv7:GESTALT, clmc2:EGFP*) were generated in our lab using Addgene plasmid #108871 and #108870[17]; Tg(*bactin2:switch*) and Tg(*msgn1:Cre-ERT2*) from Australian Regenerative Medicien Institute, Monash University, Clayton; Tg*BAC(nkx3.1:Gal4)ca101*, Tg(UAS:*Cre-ERT2*)ca105 and Tg(*Ubi*:Switch) were previously reported[15,38] and maintained in Associate Professor Huang's lab, Cumming School of Medicine, University of Calgary.

### Imaging
Epifluorescent images were acquired using a Nikon Eclipse 80i microscope equipped with a Nikon Qi2 camera. Confocal images were acquired using a Zeiss LSM 710 inverted confocal microscope. Images were processed using Fiji (ImageJ) and Adobe Photoshop (Levels function). Levels adjustments were applied independently to individual channels, where needed, to enhance visibility. All adjustments were applied uniformly across the entire image and equally to experimental and control samples. No deconvolution, gamma correction or filtering, or 3D reconstruction was applied.

### Cre-mediated lineage tracing
For *msgn1:Cre-ERT2; βactin2:Switch* embryos, tamoxifen treatment was carried out at 50% epiboly, 3 somite, or 10 somite stages using 10 μM

4-hydroxytamoxifen (4-OHT) for 24 hours. For Tg(*nkx3.1:Gal4; UAS:Cre-ERT2; ubi:Switch*) embryos, tamoxifen treatment was administered from 1 to 2 dpf with 10 μM 4-OHT. After treatment, embryos were washed three times and recovered in fish water for subsequent analysis at the appropriate stages.

### Cryosectioning
Zebrafish larvae and kidneys were fixed in 4% PFA overnight at 4 °C and washed twice in PBS and then placed in 1% low melting point agarose containing 5% sucrose and 0.9% agar (made up in water) in cryomolds and placed in 30% Sucrose/PBS cryoprotectant solution and incubated overnight at 4 °C. Cryoblocks were flash frozen on dry ice and immediately sectioned. 14 μm sections were cut in a cryostat machine (Leica SM-3050-S) with chamber temperature set to −25 °C and objective chamber set to −22 °C, and transferred onto Superfrost plus microscope slides.

**In situ hybridization and immunohistochemistry.** Both in situ hybridization and immunohistochemistry were performed on cryosectioned samples.

In situ hybridization was performed as previously described (http://zfin.org/ZFIN/Methods/ThisseProtocol.html) with the sample section slides kept in the humid chamber to prevent drying during the reaction. Antisense probes were synthesized using T3 polymerase from cDNA synthesized from kidneys of 1.5-month-old juvenile fish. To perform antibody staining, tissue sections were rinsed twice in PBS containing 0.05% Tween20 (PBST), then incubated in PBS containing 0.5% Triton X100 for 20 min to permeabilise the sections. Sections were then blocked in PBST containing 3% BSA and 5% Goat serum for at least 1 h.

Antibody staining was performed as described (https://zfin.atlassian.net/wiki/spaces/prot/pages/379420760/Antibody+staining+on+Sections), using antibodies anti Pax2 (1:500, Covance, PRB-276P-200), anti Hnf1b (1:500, Sigma, HPA002083-100UL), anti GFP (1:500, Novus, NOVNB600597), anti mCherry (1:500, Abcam, ab125096), anti Podxl (1:250, made in House by Dr. Hidetake Hurihara, Juntendo University). Sections were washed three times in PBST before the Donkey anti-mouse/Rabbit IgG (H + L) Highly cross-adsorbed Secondary antibodies (1:500, Thermo Fisher Scientific). Antigen retrieval was required for the anti-mCherry antibody using citrate solution (10 mM citric acid, 0.05% Tween, pH6.0) in 95 °C water bath for 20 min.

### Somite transplantation
The transplantation method was adapted from a previously published work[48], with several modifications made. In brief, individual donor somites were harvested from 12 or 18 somites stage β-actin-GFP

transgenic zebrafish embryos. The dechorionated donor embryos were kept in L15 media for removal of the head and yolk by using two 29 G insulin needles. The remaining trunk of the embryo was digested in 1% collagenase (Sigma) until the entire somites blocks started to detach from the embryo. The digested somites blocks were then washed by serial of L15 media supplemented with 10% FBS. Individual somite blocks were separated and selected under a fluorescent dissection microscope (Leica MZ F10) by using an eyelash and transferred to ice-cold fresh L15 media until transplantation. Dechorionated wild-type recipient embryos (12 or 18 somites), were embedded in 60 mm Petri dish with 1% low-melting/normal agarose gel (50:50) and the embryos oriented that the lateral surfaces of the targeted somite regions were at the surface of the embedding gel. Approximately 2 mL of L15 media supplemented with 10% FBS was added to the petri dish to ensure the surface was fully covered and embryos would not dry out. Under a dissecting microscope, the epidermal layer of the target somite was cut open using two Micro-Needles (0.12 mm in diameter, Ted Pella), the target somite was then hooked out using a blunt end eye lash. Immediately following the surgery, the designated donor somite was transplanted into the extirpated region of the recipient host. After transplantation, the embryos were left undisturbed and immersed in the L15 media (supplemented with 10% FBS) for at least 30 min before transferring to E3 medium. The embryos were incubated at 28 °C for recovery, survival of the embryos was checked at 1 day post transplantation. The surviving embryos were allowed to grow to 1.5 months and analyzed under fluorescent microscope. Kidney tissue was harvested and analyzed by immunohistochemistry.

### Early and late GESTALT barcode editing
The editing was performed as previously described[49], minor changes were made to fit with the experimental design of this project. In brief, sgRNAs specific to sites 1-4 of the GESTALT barcode were generated by in vitro transcription using EnGen® sgRNA Synthesis Kit, *S. pyogenes*. Tg(*hspDRv7:GESTALT, clmc2:EGFP*) F1 transgenic adults confirmed with single copy of GESTALT barcode were crossed to heat-shock inducible Cas9 F1 transgenic adults. One-cell embryos were injected with 1.5 nl of Cas9 protein (8 μM EnGen® Spy Cas9 NLS, NEB) and sgRNAs 1-4 (100 nl/μl) in salt solution (NEBuffer™ r3.1) with 0.05% phenol red to perform early GESTALT barcode editing. Injected embryos were first screened for GFP heart expression at 30 hpf to select the embryos with the presence of GESTALT barcode. These embryos were then heat-shocked for 45 min at 37 °C to induce endogenous Cas9 expression, which performed GESTALT barcode editing at sites 5-9 (late editing).

### Nuclei extraction for snRNA-Seq
Single nuclei isolation from tissue was adapted from a previously published work[50], minor changes were made to fit with the size of zebrafish tissue. Nuclei were isolated with Nuclei EZ Lysis buffer (NUC-101, Sigma) supplemented with protease inhibitor (5892791001, Sigma) and RNase inhibitors (N2615, Promega; AM2696, Life Technologies). Zebrafish brain, muscle and kidney tissue were dissected and quickly rinsed with ice-cold PBS before snap-froze. Samples were minced into <2 mm pieces with 500 μL of ice-cold NLB buffer (10 mL Nuclei EZ lysis buffer supplemented with 1 tablet of protease inhibitor and 10 μl of each RNase inhibitors) and incubated on ice for 5 min with an additional 500 μl of NLB. The tissue and NLB buffer mix was then transferred to the Dounce tissue grinder (D8938, Sigma), gently grind 30 times with Pestle A and 15 times with Pestle B. The homogenate was then incubated on ice for an additional 5 min. The homogenate was filtered through a mini-40 μm and a subsequent mini 20 μm cell strainer (43-10040-50 & 43-10020-50; PluriSelect) and then centrifuged at 500 x *g* for 5 min at 4 °C. The pellet was resuspended and washed twice with 1 mL of NLB buffer to remove residue cytoplasmic RNAs. After another centrifugation, the nuclei pellet was resuspended in Nuclei Suspension Buffer (1XPBS, supplemented with 2% BSA and 2 μl

Promega RNase inhibitor), filtered through a mini-10-μm cell strainer (43-10010-50, PluriSelect). Nuclei were counted on hemocytometers (KIMA211710, Vetriplast-10).

### Single-nucleus library preparation and sequencing
The quantified brain and muscle nuclei were pooled together in 1:2 ratio, kidney nuclei were processed on its own. The nuclei were partitioned into each droplet with a barcoded gel bead using the 10X Chromium instrument with the Chromium Single Cell 3′ Reagent Kit v.3.1 to generate transcriptome libraries. The libraries were sequenced on the Nova-Seq 6000 (Illumina).

### snGESTALT library preparation
To generate snGESTALT libraries, half of each 10X transcriptome libraries was enriched using in a two-step nested PCR reaction involving: 1) GP6 and READ1 primers and Q5 polymerase (NEB), and 2) READ1 and GREAD2 primers and Q5 polymerase. The first Q5 reaction (98 °C, 30 s, 61 °C, 25 s; 72 °C, 30 s; 15 cycles) was cleaned up with 0.6X AMPure beads and eluted in 20 μl Ultrapure water (Invitrogen); The second Q5 reaction (98 °C, 30 s, 61 °C, 25 s; 72 °C, 30 s; 8 cycles) was also cleaned up with 0.6X AMPure beads and eluted in 20 μl Ultrapure water (Invitrogen). The concentration of the PCR products was assessed using both High Sensitivity D1000 Screen tape on TapeStation (Aglient) and Qubit™ dsDNA HS Assay Kit. Final PCR was then carried out to incorporate sequencing adapters and sample indexes as described in the Chromium Single Cell 3′Reagent kit v.3.1. The NEB-Next Library Quant kit was used for library quantification. Libraries were sequenced using MiSeq (300 cycles kit) and 20% PhiX spike-in. Sequencing parameters: Read1 28 cycles and Read2 300 cycles.

### Sequences of Oligonucleotides
GP6: 5′-GAGGACTACACCATCGTGGAG
  READ1: 5′-CTACACGACGCTCTTCCGATCT;
  GREAD2: 5′-GTGACTGGAGTTCAGACGTGTGCTCTTCCGATCTCACCTGTTCCTGTAGAAATC;

### Processing of raw transcriptome sequencing reads
In brief, sequencing data were demultiplexed and aligned to the zebrafish genome (GRCz10) using Cell Ranger v6.0.1 with the setting `--include-introns`. An expression matrix was generated based on UMI counts per gene per cell.

### snRNA-Seq data analysis
Seurat v4.0[51] was used for downstream analysis. We analysed each zebrafish separately and removed low quality nuclei with less than 200 genes detected. We also excluded nuclei with a high percentage of UMIs mapped to mitochondrial genes (>0.02). An R package, SoupX v1.4.5[52], was used to remove ambient RNA. Briefly, ambient RNA expression is estimated from the empty droplets. The contamination fraction is then calculated with the setting "autoEstCont" followed by the "adjustCounts" function with default parameters to generate a UMI count matrix. We also used scDblFinder v1.4.0[53] to remove doublets. Data were normalised and scaled using SCTransform[54] function. Mitochondria gene expression was removed from the datasets. We integrated four brain and muscle datasets, and three kidney datasets using integration anchors[55]. We then performed dimension reduction, clustering and differential expression analysis on individual clusters using the FindAllMarkers function in Seurat. Canonical marker genes together with the top differentially expressed genes for each cluster were used to identify cell type populations. Pathway analysis was performed using ClusterProfiler v3.1.8[56].

### GESTALT barcode analysis
Sequencing data from GESTALT libraries were processed with the GESTALT pipeline (https://github.com/mckennalab/SingleCellLineage).

In brief, read 1 and read 2 were concatenated to generate GESTALT barcode reads with the 10X cell barcodes. A consensus sequence was called by merging all reads that shared the same cell barcode. Consensus sequences were then aligned to a reference sequence using NEEDLEALL aligner with the default settings except `-primersToCheck=FORWARD` and `-umiLength=28`. The GESTALT barcodes were matched with the corresponding nuclei from the transcriptome datasets based on the 10X cell barcode. Nuclei with <2 GESTALT barcode UMIs were excluded from the lineage trees. Lineage trees were constructed using TreeUtils (https://github.com/mckennalab/TreeUtils) with the setting `--subsetFirstX 4` to anchor the tree with the early edit sites (site 1-4) and extended it with the late edit sites (site 5-9). The trees were then annotated with tissue type.

### Reporting summary

Further information on research design is available in the Nature Portfolio Reporting Summary linked to this article.

### Data availability

The snRNAseq data generated in this study have been deposited in the Gene Expression Omnibus (GEO) database under accession code GSE286280. Source data for Fig. 4o are provided with this paper. Source data are provided with this paper.

### Code availability

All snRNA-seq data analyses were performed using standard protocols with previously described R packages[51]. GESTALT data were processed using Single-cell GESTALT pipeline available at GitHub (https://github.com/mckennalab/SingleCellLineage).

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

## Acknowledgements

We thank Prof Alexander F Schier, the Director of the Biozentrum University of Basel, Switzerland, and Dr Bushra Raj from the Department of Molecular and Cellular Biology, Harvard University, Cambridge Massachusetts, USA, for their technical support on the GESTALT experiment and data analysis. We also thank Dr. Nikki Freed and Dr. Jieyun Wu from the sequencing facility, Auckland Genomics, the University of Auckland, for technical assistance with snRNA-seq and GESTALT library sequencing. This study was supported by grant 18-UOA-151 from the Marsden Fund, Royal Society of New Zealand (A.J.D.), NHMRC Australia Investigator Grant 2016338 (P.D.C.), Canadian Institutes of Health Research PJT-169113 (P.H.) and NIH R01DK121014 (L.L.O.).

## Author contributions

Conceptualization, Z.P. and A.J.D.; snRNA-Seq analysis, Z.P. and T.V.; somite transplantation, Z.P.; lineage tracing, Z.P., P.D.N., and H.G.C.; visualization, Z.P., T.V., and A.J.D.; formal analysis, Z.P. and A.J.D.; resources, P.D.N., P.T.D., K.K., and P.H.; writing – original draft, Z.P. and A.J.D.; writing – review and editing, P.H., P.D.C., and L.L.O.; supervision, A.J.D.; project administration, Z.P. and A.J.D.; funding acquisition, Z.P. and A.J.D.

## Competing interests

The authors declare no competing interests.
