## [Peer Review file · Nature Communications]

Somites are a source of nephron progenitors in zebrafish

Corresponding Author: Dr Alan Davidson

Version 0:

Reviewer comments:

Reviewer #1

(Remarks to the Author)

Key results

In this manuscript by Peng and colleagues the authors determine that the somites are a source of nephron progenitors in zebrafish. This is important, and surprising, because up until now it was thought that the intermediate mesoderm is the sole source of the nephron progenitor population across vertebrates. These findings may indicate that the somite to kidney pathway they discover here, may be ancestral. The authors use three powerful and complementary approaches to test their model. Multiple methods arriving at the same conclusion, contributes to the high rigor of this study. However, I have many comments, mostly relating to poor presentation, that are intended to help the authors improving clarity and context which is not currently acceptable. Overall I am enthusiastic about this study and it's findings, but was disappointed at the numerous presentation problems. Addressing these distracting issues will help to propel this manuscript into a top tier journal like Nature Communications.

Validity

As stated above the diverse approaches leading to the same conclusion make the validity and robustness strong. Impressively, the authors also clearly issue some caveats associated with each method further demonstrating rigor. There are other cases where additional caveats are warranted and omitted from the current draft. For example, regarding the transplantation can the authors be sure they are transplanting JUST somites and there are not other cells that are coming along for the ride that then contribute to the kidney? This is a very important consideration for their conclusions. Also regarding the transplantations, do the authors think that just the anterior somites are competent to contribute to the kidney. If they transplanted a posterior donor somite into an anterior position in the host, would they still detect somite contribution to the kidney.

Significance

These findings are potentially significant because a better understanding of kidney development in vertebrates, beyond mammals and birds, provides a more complete picture and may eventually lead to insights informing regenerative therapies. However, there are a few places where the rationale doesn't quite add up. For example, on line 54 the authors hypothesize that the NPCs come from the somites, but this is not thoroughly set up. Why would the authors hypothesize this? Just because they appear at the ventral border of anterior somites? I'm not fully convinced that location alone warrants the hypothesis.

Data and methodology

The presentation quality is a major weakness of this manuscript. Many figures do not have appropriate labels. For example, in Fig. 1A the blue label is not indicated. There are many other instances of similar poor labeling that I do not detail but the authors need to carefully edit their figures and legends for completion.

The methods are also lacking. For example, the details of the tamoxifen treatment (concentration and timing) are not easily found and are important for reproducibility. They do not describe the $nkx3.1:Gal4$ line in the methods. While they do discuss to generation of this line in the text, they state that the transgene consists of the $nkx3.1$ promoter which while technically true, does not accurately describe this line since this transgene is a 144KB BAC containing many other (presumed regulatory sequences) not JUST the promoter. Finally, the authors call this line $nkx3.1-cre$ which is not correct. The authors can define the use of $Gal4/UAS$ and how that is used to drive cre , but this must be defined at the first instance. Transgene colon/dash usage is overall incorrect and inconsistent.

There are many cases where abbreviations are in the figure but not defined in the legend. This can be frustrating to the reader and need to be fixed.

Gene nomenclature needs to be properly followed throughout. For example in Fig. 1 G gene names are not italicized when they should be per zebrafish convention. This is not just cosmetic because proper gene nomenclature allows the reader to know if the author is referring to genes or gene product and from what species.

Regarding the GESTALT study, please describe why the muscle and brain were pooled together. It is fascinating that there is contralateral contribution from the donor.

Clarity and context

There are many cases in the text where that authors assume too much of the reader especially as it relates to different tissues and developmental processes that need to be better explained in the introduction so that the reader can appreciate the important contributions in this paper. Some examples include: I was not satisfied with the description of the IM and what is known about that tissue and it's derivatives. The authors do not describe the myotome until the discussion. Rather, they just say "... mCherry fibers were also observed in the ventral part of myotome". This feels like it is missing a word but more importantly, the myotome has not yet been introduced. Do not assume your reader knows what this is. The same thing is true for the sclerotome, it comes out of nowhere and there needs to be a more complete description of the various zebrafish structures that are derived from the sclerotome. Similarly, on line 189 the authors say "pax2 positive nephrons" but we have not yet been introduced to pax2 and why it's important. In a similar vein there needs to be a better description in the introduction of all the different parts of the kidney. The authors jump into the pronephros, mesonephros, metanephros, pronephric tubules without properly introducing any of these structures. This is important contexts that are omitted. Don't force your reader to resort to google searches in order to gain the information required to understand your study. Other examples of poor clarity abound, including Extended data figure 4 the authors show an image that they report is 50% epiboly, but the animal appears far older. Is this a mistake or am I not understanding? In this same figure, what is this marker in H? What are the abbreviations in G? I found myself getting increasingly frustrated with these sloppy omissions. In Figure 2 all the abbreviations need to be defined in the legend. There are many, many cases like this so I won't belabor the point further but presentations is in dire need of improvement. Finally, the entire manuscript needs careful proofreading there are many incomplete sentence fragments that are very distracting (line 155-6 is just one example). When the authors report "length" of the fish (line 66), are they referring to "standard length" per convention?

Reviewer #2

(Remarks to the Author)

Peng et al. provide convincing evidence that some nephrons can derive from somites and thus establish a previously underrecognized source of the adult kidney. The paper is impressive because it uses three independent and complimentary fate mapping approaches.

Specific comments.

1. The clustering analysis is nice, but it does not directly address the lineage relationship that is the focus of the study. This part could be shortened in the main text and put into supplement.
2. The snGESTALT experiment is intriguing, but I would like to understand more clearly how the shared lineages are related. Are you sure that the barcode editing that generated Fish 1 branch a2 and Fish 2 branch b1 happened during later not earlier editing? It would be trivial if there is a common progenitor during blastula stage. How sure are you that the common progenitor was edited after the 12-somite stage and resided in a somite?
3. How sure are you that only somite 3 was transplanted without any neighboring material?

Reviewer #3

(Remarks to the Author)

This study challenges the long-standing view that vertebrate nephron progenitors arise solely from the intermediate mesoderm. Using zebrafish as a model, the authors apply three complementary lineage-tracing strategies—snGESTALT, somite transplantation, and Cre-lox fate mapping—to demonstrate that somites contribute to the developing mesonephros. These findings uncover a previously unrecognized source of nephron progenitors and suggest a potential role for ventral sclerotome- or myotome-derived muscle stem cells. Overall, the data are compelling and support a significant revision of nephron lineage origin. However, it remains unclear whether somites also contribute to the renal stroma. Clarifying this point would further strengthen the impact of the study.

Major points:

Stroma

It would be helpful to address if somites can contribute to renal stroma. It appears that some stroma cells were labeled with GFP in somite transplantation (Fig3) and with mCherry in lineage tracing (Fig4). Some co-staining with stromal markers will clarify this issue. Furthermore, snGESTALT analysis may show that renal stromal cells also share the same barcodes with muscle cells, increasing the number of barcode sharing events.

snGESTALT

The color coding of black and red dots is not consistent between Fig2 and Extended Data Fig2. For Fish2, b1 is red in Fig2 and black in Extended Data Fig2. If b1 represents early edits as indicated in Extended Data Fig2, it may be a coincidence that kidney and muscle cells share the same barcode. It is unclear how early edits can be distinguished from late edits.

Minor points:

Relative role of somites vs. intermediate mesoderm

It's unclear whether somites are the sole source of mesonephric progenitors or if they contribute alongside the intermediate mesoderm. Clarifying this point would help position the findings within existing developmental models.

Fig1b

It's unclear why barcode editing needed to be done in two different stages. The purpose of early editing with gRNA 1-4 in a single cell embryo is unclear. This should be explained better.

Extended Data Figure 2

UMAPs appear identical for Fish1 and Fish2. The description for UMAPs is absent in the figure legend.

Many abbreviations were not defined in the figure legend, making it difficult to read. Too many to list them all.

Stages were indicated by either hdf, dpf or somite stage, making it difficult to access their relative timing.

What is the difference between RPC and NPC? If they refer to the same cells, please choose one and use it consistently throughout the manuscript.

Below sentence needs at least one relevant reference (page=3, lines 39-42):

However, zebrafish kidney development challenges this model. Compared to the mouse kidney development, the zebrafish pronephros forms early from the entirety of the IM, while the mesonephros develops after a significant temporal gap from nephron 42 progenitor cells (NPCs) whose origin is unclear.

According to the reference below, these are not an exact word to mention different muscle cell types or muscle fibers. (page:5, lines 87, muscle cell types (fast and slow)). Please replace it with a better reference.

Authier F. J. (2000). Classification des cellules musculaires [Classification of muscle cells]. *Revue des maladies respiratoires*, 17(2 Pt 2), 525–530.

Typo

(Page: 23, line 477, cryomoulds)

Please add at least one reference that showed Foxd1 expression in somite. (page=12, line 257):

Lineage tracing of mouse stromal precursors using Foxd1-cre, which shows earlier expression in the somites, labels the metanephric stroma but also small populations of metanephric NPCs [35].

Version 1:

Reviewer comments:

Reviewer #1

(Remarks to the Author)

The authors have sufficiently responded to all our critiques. This work represents a nice contribution to the field and furthers our understanding of the developmental and evolutionary origins of the kidney.

Reviewer #2

(Remarks to the Author)

The authors have addressed my concerns.

Reviewer #3

(Remarks to the Author)

The authors addressed my concerns reasonably well.

I believe the data presented here support the authors' conclusions, but I still have the following minor comments.

1. Gene names should be italic (Fig1d, e, h; Extended Data Figure 1f, h, k)
2. Extended Data Figure 2: I suspect that the white bars that represent the full barcode might not be positioned correctly because (1) the second half of white bars show no edits, and (2) some edits are outside of the white bars. Adding a bar with

nine barcode sites might be helpful to distinguish early edits from late edits.

Thank you for giving us the opportunity to revise our manuscript titled "Somites are a source of nephron progenitors in zebrafish" for publication in *Nature Communications*. We appreciate the valuable feedback provided by you and the reviewers. Below, we have addressed each comment point-by-point.

Reviewer 1:

1. *Comment: For example, regarding the transplantation can the authors be sure they are transplanting JUST somites and there are not other cells that are coming along for the ride that then contribute to the kidney?*

Response: The *mshn-cre* and *nkx3.1* lineage labelling experiments provide good evidence that the somitic lineage is a source of the NPCs. However, you raise a good point about whether contaminating (non-somitic) cells attached to the somite may then contribute to the kidney. The best candidate for these would be the adjacent IM. When we isolate the somites it's very easy to strip away the IM so we are confident that there aren't contaminating IM cells. However, to add more weight to this we have performed additional experiments where we have transplanted somites from donors carrying the *cdh17:EGFP* transgene, which is expressed in the IM/pronephric tubules and will indicate if any contaminating IM is 'coming along for the ride'. From 23 rounds of transplantation, we did not see any GFP⁺ pronephric tissue in the recipient fish, adding weight to our sense that the somites are 'clean'. We've added this data as Extended Data Figure 3. i-j and as the text below in the results section:

"Despite the somite being easily separable from the adjacent IM, we explored the potential for contaminating IM cells to be co-isolated. To do this, we performed somite transplantations with *Tg(cdh17:EGFP)* donors, which is expressed by the IM/pronephric tubules, but found no donor-derived GFP⁺ kidney cells in the recipient fish at 4 dpf (n=23; Extended Data Figure 3i,j)."

2. *Comment: Also regarding the transplantations, do the authors think that just the anterior somites are competent to contribute to the kidney. If they transplanted a posterior donor somite into an anterior position in the host, would they still detect somite contribution to the kidney.*

Response: This is a great question and initially we thought we could test all the somites along the AP axis for their nephron-forming competency. However, somite transplantation experiments are just too technically difficult, time-consuming and inefficient. We haven't considered the idea of moving a more posterior somite into an anterior position, that would be interesting to explore. In the Sandkhol Carp paper, they reported presumptive NPCs appearing all down the trunk at quite posterior somite positions (at least as far as somite 21), so our speculation is that most (all?) of the trunk somites are competent to form NPCs.

3. *Comment: For example, on line 54 the authors hypothesize that the NPCs come from the somites, but this is not thoroughly set up. Why would the authors hypothesize this? Just because they appear at the ventral border of anterior somites? I'm not fully convinced that location alone warrants the hypothesis.*

Response: Apologies for not setting this up more clearly in the introduction. The hypothesis is borne from the Sandkhol carp work and our observation that we see *Lhx1a:EGFP*⁺ NPCs first appearing next to the somite. We have re-worked the introduction and beginning of the results with the following text to make this more clear:

Introduction: “Indeed, a careful histological analysis of mesonephros formation in Sandkhol Carp larvae suggests that presumptive NPCs first originate near the base of the somites— blocks of paraxial mesoderm best known for their contributions to skeletal muscle, dermis, and vertebrae^[8-11]”

Beginning of Results: “Consistent with the prior observation in Sandkhol Carp, we found that zebrafish NPCs, which are fluorescently labelled in the *Tg(lhx1a:EGFP)* transgenic line^[7,16], first appear at the ventromedial border of the anteriormost somites, adjacent to the pronephric tubules (Figure 1a). From here they migrate and expand in number around the axial vessels (Extended Data Figure 1a,b)^[7].

To explore the hypothesis that these NPCs originate from the somites, we performed a lineage mapping analysis using a modified version of the scGESTALT...”

4. *Comment:* The methods are also lacking.

Response: Apologies for this-- we have made improvements to the Methods section to provide a more detailed and comprehensive description. We have added the methods for the tamoxifen treatments:

“Cre-mediated lineage tracing. For *msgn1:Cre-ERT2; βactin2:Switch* embryos, tamoxifen treatment was carried out at 50% epiboly, 3 somite, or 10 somite stages using 10 μM 4-hydroxytamoxifen (4-OHT) for 24 hours. For *Tg(nkx3.1:Gal4; UAS:Cre-ERT2; ubi:Switch)* embryos, tamoxifen treatment was administered from 1 to 2 dpf with 10 μM 4-OHT. After treatment, embryos were washed three times and recovered in fish water for subsequent analysis at the appropriate stages. Page 25 – Method section”

5. *Comment:* in Fig. 1A the blue label is not indicated

Response: We have now corrected the label and added ‘DAPI’ to this figure.

6. *Comment:* They do not describe the *nkx3.1:Gal4* line in the methods. While they do discuss to generation of this line in the text, they state that the transgene consists of the *nkx3.1* promoter which while technically true, does not accurately describe this line since this transgene is a 144KB BAC containing many other (presumed regulatory sequences) not JUST the promoter. Finally, the authors call this line *nkx3.1-cre* which is not correct. The authors can define the use of *Gal4/UAS* and how that is used to drive *cre*, but this must be defined at the first instance.

Response: We thank the reviewer for pointing this out. We have updated the Methods section to include *TgBAC(nkx3.1:Gal4)ca101*, *Tg(UAS:Cre-ERT2)ca105* and *Tg(Ubi:Switch)*. We appreciate the reviewer’s observation and agree that our previous shorthand referring to this line as ‘*nkx3.1-cre*’ was imprecise. To clarify, the correct terminology is *nkx3.1:Gal4; UAS:Cre-ERT2*. We have now defined the use of the *Gal4/UAS* system and its application to drive Cre recombinase in *nkx3.1*-expressing cells at the first instance where this is mentioned in the manuscript:

“...we conducted conventional 4-hydroxytamoxifen (4-OHT)-inducible Cre-mediated (CreERT2) genetic fate-mapping experiments *in vivo* using transgenic lines employing the paraxial mesoderm-specific *mesogenin1* (*msgn1*) promoter^[34] and a GAL4-UAS^[35] system involving the sclerotome marker *NK3 homeobox 1* (*nkx3.1*)^[15] (Figure 4a)....

Similarly, in *nkx3.1:Gal4; UAS:Cre-ERT2; ubi:Switch* fish, *nkx3.1*-expressing sclerotome cells were targeted for Cre-mediated recombination at 24 hpf (Figure 4b).

This approach enabled us to track and compare the fate of cells derived from the paraxial mesoderm (*msgn1*) at early stages and the sclerotome subpopulation (*nkx3.1*) at later stages.”

7. *Comment: There are many cases where abbreviations are in the figure but not defined in the legend.*

Response: Thank you for pointing this out. We have carefully reviewed all the figures and ensured that all abbreviations are now clearly defined in the corresponding legends. The updated legends can be found on pages [17-24]

8. *Comment: Gene nomenclature needs to be properly followed throughout. For example in Fig. 1 G gene names are not italicized when they should be per zebrafish convention.*

Response: Thank you for pointing this out. We have italicized the gene names in Fig.1g.

9. *Comment: Regarding the GESTALT study, please describe why the muscle and brain were pooled together.*

Response: We pooled the muscle and brain samples due to the high cost of the snGESTALT experiment (~\$80k). Muscle and brain tissues exhibit very distinct gene expression profiles, so despite pooling, we can confidently deconvolute the cell populations based on their gene profiles.

10. *Comment: There are many cases in the text where that authors assume too much of the reader especially as it relates to different tissues and developmental processes that need to be better explained in the introduction so that the reader can appreciate the important contributions in this paper.*

Response: We have taken this on board and have revised the introduction to provide a more comprehensive explanation of the different tissues and developmental processes. Specifically, we have:

- a. Expanded background information on IM, different parts of kidney: pronephros, mesonephros, metanephros.
- b. Added a paragraph on somites and its derivatives
- c. Clarified key concepts: *Hnf1b*, *pax2a*

For instance in the introduction we had added the following:

“In the mouse, the IM forms a relatively dense cord of cells with distinct subsets developing into the nephric duct, mesonephros and metanephros, within a short time period (2-3 days)^[1]. By contrast, the zebrafish IM comprises a sparse, narrow strip of cells that appears to differentiate in its entirety into the pronephros[6]. Within 24 hours post-fertilization (hpf), all the major pronephric cell types have arisen and by 48 hpf filtration begins. This rapid development likely reflects the physiological need for the zebrafish embryo to osmoregulate in an aquatic environment outside of the maternal body. The zebrafish mesonephros arises much later, around 10 days post-fertilization (dpf), and this has been traced to the appearance of individual nephron progenitor cells

(NPCs)[7, 8]. These cells first appear posterior to the swim bladder and then migrate onto the pronephros, cluster together, and proliferate into nephrons[7, 8]....

The somites, segmented blocks of paraxial mesoderm that become compartmentalised into three distinct regions: the myotome, which forms skeletal muscles; the dermomyotome, which generates the dermis and muscle; and the sclerotome, which gives rise to the vertebrae and rib cartilage. These compartments have been extensively studied for their contributions to musculoskeletal development [9-11]. However, lineage tracing studies have revealed that somites are a diverse source of different progenitor cell types, far exceeding their classical designation as musculoskeletal precursors. It is now appreciated that somitic cells can also give rise to brown adipose tissue[14], endothelial cells[15, 16] and fibroblasts[17], indicating a developmental potential much broader than previously appreciated.”

11. *Comment: Extended data figure 4 the authors show an image that they report is 50% epiboly, but the animal appears far older. Is this a mistake or am I not understanding?*

Response: Thank you for catching this. The label was indeed misleading, and we have corrected the figure label and related figure legend. To clarify, the tamoxifen treatment was performed at the 50% epiboly, 3-somite, and 10-somite stages. The larvae were imaged at 48 hpf.

12. *Comment: When the authors report “length” of the fish (line 66), are they referring to “standard length” per convention?*

Response: Yes we are referring to standard length: snout to the base of the caudal fin. To make this clear we have inserted into the text “Standard Length” where this first appears.

Reviewer 2:

1. *Comment: The clustering analysis is nice, but it does not directly address the lineage relationship that is the focus of the study. This part could be shortened in the main text and put into supplement.*

Response: We appreciate the reviewer's feedback and we have gone back and forth with how we could shorten this section while also retaining the information that is needed for later sections. While we understand the concern about not directly addressing the lineage relationship, we feel the analysis, as written, is essential for identifying and defining the cell populations for the downstream barcode analysis, which is crucial for establishing the lineage relationship.

2. *Comment: Are you sure that the barcode editing that generated Fish 1 branch a2 and Fish 2 branch b1 happened during later not earlier editing? It would be trivial if there is a common progenitor during blastula stage. How sure are you that the common progenitor was edited after the 12-somite stage and resided in a somite?*

Response: Technically, early edits can only be introduced during the early blastula stage as we injected the 4 gRNAs that specifically target the first 4 barcodes at the 1-cell stage. To ensure the specificity of early editing, we incubated the Cas9-gRNA complex for 15 minutes prior to injection, which was necessary to ensure the efficient formation of the Cas9-gRNA complex. Additionally, the original Gestalt paper [Raj et al., 2018] has demonstrated that Cas9-gRNA-mediated editing is specific to the first four barcode sites at these early stages. We have added more clear details on our barcode editing strategy in the main text, on page 5-6, line 100-105.

“This approach maps developmental relationships between different lineages of cells using a combination of genetic barcode editing by CRISPR/Cas9 and single cell RNA-Sequencing. Transgenic zebrafish carrying the heat shock inducible GESTALT barcode (sites 1-9) were crossed to a heat shock-inducible Cas9 line (constitutively expressing guide RNAs to sites 5-9) and injected at the single-cell stage with Cas9 protein and guide RNAs 1-4. This ensures editing occurs at barcode sites 1-4 during the pre-gastrulation stages, capturing the earliest lineage decisions (Figure 1b). The embryos were then heat shocked at the 12-somite stage (15 hpf) to induce a second round of editing at sites 5-9 during mid-somitogenesis (Figure 1b). This dual editing strategy allows us to capture both early lineage bifurcations and later developmental decisions, enabling reconstruction of more detailed lineage trees.”

To validate the timing and efficiency of the editing, we followed the published protocols from Raj et al. (2018, *Nature Protocols*) for the injection and heat-shock procedure. We tested early, late, and double edits and analyzed the genomic DNA (gDNA) from fin-clipped larvae by PCR. If no editing occurred, we observed a solid band at ~300 bp. Early edits showed a faint smear between 200–300 bp, with a major band at 300 bp, and double edits produced a larger smear across the 100–300 bp range (see figure below). These findings match the results reported in Figure 5 of the Raj et al. *Nature Protocols* publication. Furthermore, we performed miSeq genomic DNA sequencing to assess the barcode editing efficiency, with results presented in Extended Data Figure 1c. The two references mentioned above are:

1. Raj B, Gagnon JA, Schier AF. Large-scale reconstruction of cell lineages using single-cell readout of transcriptomes and CRISPR-Cas9 barcodes by scGESTALT. *Nat Protoc.* 2018 Nov;13(11):2685-2713. doi: 10.1038/s41596-018-0058-x. PMID: 30353175; PMCID: PMC6279253.
2. Raj, B., Wagner, D., McKenna, A. et al. Simultaneous single-cell profiling of lineages and cell types in the vertebrate brain. *Nat Biotechnol* 36, 442–450 (2018). <https://doi.org/10.1038/nbt.4103>

As further support for the timing of the edits, if we had targeted a common progenitor during the blastula stage, we would expect to observe a shared lineage across tissues (brain, muscle, kidney) and a higher number of matched cells, as there are relatively few blastomeres that can be edited at this stage. However, our data did not show such patterns, which suggests that the common progenitor was edited after the blastula stage, likely after the 12-somite stage, as we intended.

3. *Comment: How sure are you that only somite 3 was transplanted without any neighboring material?*

Response: Please see the response to Reviewer 1.

Reviewer 3:

1. *Comment: It would be helpful to address if somites can contribute to renal stroma. ... snGESTALT analysis may show that renal stromal cells also share the same barcodes with muscle cells, increasing the number of barcode sharing events.*

Response: The difficulties with analysing the stroma is that (i) they haven't been well characterized so there are not any validated markers and (ii) the zebrafish kidney is also the site of haematopoiesis, therefore there are a large number of different interstitial, stromal, and niche cells amongst the renal tubules. We do see non-tubular mCherry⁺ cells in the *nkx3.1*-derived lineage and have added this data to Extended Data Figure 4i-j and text to page 11-12, line 245-248

“In addition to the nephron segments, we have also observed contribution to stromal-like cells in the kidney. However, in the absence of validated stromal markers in zebrafish, the identities of these cells are unclear (Extended data figure 4i).”

2. *Comment: The color coding of black and red dots is not consistent between Fig2 and Extended Data Fig2. For Fish2, b1 is red in Fig2 and black in Extended Data Fig2. If b1 represents early edits as indicated in Extended Data Fig2, it may be a coincidence that kidney and muscle cells share the same barcode. It is unclear how early edits can be distinguished from late edits.*

Response: Thank you for picking up this mislabelling. We have updated and corrected the inconsistency in both Figure 2 and Extended Data Figure 2 and updated the text in the manuscript and figure legend. The original ‘Clade b’ in Extended data figure 2 should be ‘unedited’ and has been relabelled. The lineage previously labelled as ‘b1’ is now designated ‘Clade b’, and ‘b2’ is renamed ‘Clade c’. In all cases, black nodes represent early barcode edits at sites 1–4 (introduced before gastrulation), and red nodes represent late edits at sites 5–9 (introduced after the 12-somite stage). These distinctions are now consistently reflected in both the figures and legends.

To address the concern about how early and late edits can be distinguished, we have provided additional details on our experimental approach in the text (see response to Reviewer 1's point 2). Early barcode editing (black nodes) occurs at sites 1-4 prior to gastrulation, while later editing (red nodes) occurs at sites 5-9 following heat-shock induction at the 12-somite stage. This distinction is also clearly outlined in the Methods section, where we describe the timeline of barcode editing using the heat-shock-inducible GESTALT barcode system (Figure 1b).

3. *Comment: It's unclear whether somites are the sole source of mesonephric progenitors or if they contribute alongside the intermediate mesoderm. Clarifying this point would help position the findings within existing developmental models.*

Response: We agree that this is an important discussion point. We have added the following to the Discussion section:

“While our results link the somites to the mesonephric kidney, we cannot completely rule-out a dual contribution that also includes the IM. At present, we do not favour such a model as the

temporal gap between when the IM forms (~10 hpf) and NPCs appearing 10 days later is a substantial length of time for these cells to persist. We have conducted extensive gene expression mapping of NPC markers, including *pax2a*, *wt1b*, *lhx1a*, and have not seen any evidence of ‘remnant’ IM. It is possible that some IM cells differentiate into an intermediary cell type that does not express traditional markers of NPCs yet retains the potential to adopt an NPC fate later in development.”

4. *Comment: It is unclear why barcode editing needed to be done in two different stages. The purpose of early editing with gRNA 1-4 in a single cell embryo is unclear. This should be explained better.*

Response: We appreciate the reviewer’s insightful question. To address this, we have added more explanation on page 5-6 to clarify the purpose of both early and late editing (see response to Reviewer 1’s point 2)

5. *Comment: UMAPs appear identical for Fish1 and Fish2. The description for UMAPs is absent in the figure legend.*

Response: Apologies, we have corrected the UMAP for fish 2 and added a description for both UMAPs in the figure legend.

6. *Comment: Many abbreviations were not defined in the figure legend, making it difficult to read. Too many to list them all.*

Response: We have reviewed all figure legends and ensured that all abbreviations are now defined upon their first use. This should make the figures easier to understand.

7. *Comment: Stages were indicated by either hdf, dpf or somite stage, making it difficult to access their relative timing.*

Response: We have added the hpf equivalent when somite stage is mentioned to clarify the relative timing.

8. *Comment: What is the difference between RPC and NPC? If they refer to the same cells, please choose one and use it consistently throughout the manuscript.*

Response: Thank you for pointing out these typos (we changed our nomenclature from RPC to NPC but didn’t catch all the instances). We have now updated all instances of RPC to NPC to consistently refer to the nephron progenitor cells throughout the manuscript.

9. *Comment: Below sentence needs at least one relevant reference (page=3, lines 39-42): However, zebrafish kidney development challenges this model. Compared to the mouse kidney development, the zebrafish pronephros forms early from the entirety of the IM, while the mesonephros develops after a significant temporal gap from nephron progenitor cells (NPCs) whose origin is unclear.*

Response: Thank you for pointing this out. This block of text has now been amended and references added:

“In the mouse, the IM forms a relatively dense cord of cells with distinct subsets developing into the nephric duct, mesonephros and metanephros, within a short time period (2-3 days)^[1].

By contrast, the zebrafish IM comprises a sparse, narrow strip of cells that appears to differentiate in its entirety into the pronephros^[4]. Within 24 hours post-fertilization (hpf), all the major pronephric cell types have arisen and by 48 hpf filtration begins^[4,5]. This rapid development likely reflects the physiological need for the zebrafish embryo to osmoregulate in an aquatic environment outside of the maternal body.”

10. *Comment: According to the reference below, these are not an exact word to mention different muscle cell types or muscle fibers. (page:5, lines 87, muscle cell types (fast and slow)). Please replace it with a better reference. Authier F. J. (2000). Classification des cellules musculaires [Classification of muscle cells]. Revue des maladies respiratoires, 17(2 Pt 2), 525–530.*

Response: Unfortunately we were only able to access the abstract of the paper that the reviewer suggested, perhaps because it is written in French, but from this it appears that it refers to fiber type distinctions in mammalian muscle. The abstract states “According to morphological and functional criteria, skeletal muscle fibers are classified as type I fibers (slow-twitch oxidative), type IIA fibers (fast-twitch oxidative glycolytic) or type IIB fibers (fast-twitch glycolytic).” These distinctions are particular to mammals and are not immediately attributable physiologically and molecularly to teleost muscle as they refer specifically to the expression of specific myosins, and this is unique to mammals. Furthermore, unlike mammals, where each muscle contains a unique mix of these fiber types, in teleosts slow twitch and fast fibers are topographically separable from each other, with fast muscle forming the deeper muscle of the myotome and slow muscle found superficially to this on the outside of the myotome. While it is likely more complex than this in ontogeny it is not correct to simply utilize the mammalian distinctions as this relationship has not been established in teleosts such as zebrafish. Thus the accepted nomenclature in the literature for teleosts at this point is fast versus slow twitch fibres. A review of the subject can be found here: <https://pubmed.ncbi.nlm.nih.gov/23811405/> Jackson HE, Ingham PW. *Mech Dev.* 2013 Sep-Oct;130(9-10):447-57. doi: 10.1016/j.mod.2013.06.001

11. *Comment: Typo (Page: 23, line 477, cryomoulds)*

Response: Thanks for pointing this out. We have corrected the typo.

12. *Comment: Please add at least one reference that showed Foxd1 expression in somite. (page=12, line 257)*

Response: We have added Kobayashi et al. (2014), who showed *Foxd1* expression in renal progenitors during mammalian kidney organogenesis, and Tamplin et al that reported it in mouse somites in their supplemental data (images online at <https://www.informatics.jax.org/assay/MGI:5291697>).

Owen J. Tamplin, Brian J. Cox, Janet Rossant. Integrated microarray and ChIP analysis identifies multiple *Foxa2* dependent target genes in the notochord.